# Determination of Biological and Molecular Attributes Related to Polystyrene Microplastic-Induced Reproductive Toxicity and Its Reversibility in Male Mice

**DOI:** 10.3390/ijerph192114093

**Published:** 2022-10-28

**Authors:** Tao Liu, Baolian Hou, Yecui Zhang, Zhiping Wang

**Affiliations:** Department of Occupational and Environmental Health, School of Public Health, Cheeloo College of Medicine, Shandong University, Jinan 250012, China

**Keywords:** microplastic, mitochondria, testis, sperm, ROS

## Abstract

Microplastics exist not only in the natural environment, but also in human tissue such as blood and even placenta. Polystyrene microplastic exposure can cause abnormal sperm quality in mice; however, the mechanism is unclear, and whether sperm abnormalities can be restored has not been reported. ICR mice were exposed to 5 μm polystyrene microplastics through the drinking water. After one spermatogenic cycle, mitochondrial damage was observed to explain the possible cause of sperm damage. After 1–2 spermatogenic cycles of recovery, whether the damaged sperm could be recovered was observed. The results show that polystyrene microplastics caused a decrease in the mitochondrial membrane potential, an imbalance of kinetic homeostasis, a change in genetic characteristics, mitophagy, and a decrease in the ATP content in mouse testicular tissue. Oxidative stress may be the cause of mitochondrial damage. After 1–2 spermatogenic cycles, mitochondrial damage was restored and sperm quality was improved. This study explored the mitochondrial causes of reproductive toxicity of polystyrene microplastics and the reversibility of reproductive toxicity, providing data for further research on the toxicity of microplastics and the prevention and treatment of its harm.

## 1. Introduction

Microplastics (MPSs) are plastic particles with a particle size of less than 5mm [1]. Polystyrene plastic particles with an average diameter of 0.5 mm were first discovered in the coastal waters of New England in 1972 [2], and the term “microplastic” was first proposed by British marine biologist Professor Richard Thompson in 2004 [3]. MPSs are found not only in nature [4] but also in the human living environment.

Dutch scientists have detected microplastics in groundwater and drinking water [5], and there are many reports of microplastics being found in aquatic products and salt for human consumption [6,7,8]. For example, an observational study of commercially purchased mussels and spendophytes by Ghent University in Belgium pointed out that European shellfish consumers may consume 11,000 pieces of microplastics with diameters ranging from 5 to 25 μm per year [9]. In addition to aquatic products, microplastics have also been found in agricultural and sideline products and baby milk powder. In May 2020, scientists from the University of Catania in Italy found tiny plastic debris in fruits and vegetables when analyzing samples collected from carrots, lettuce, apples, and pears [10]. A study published in *Nature Food* in October 2020 estimated that infants may ingest up to 16 million microplastic particles per liter of infant formula prepared in polypropylene plastic bottles [11]. In addition, microplastics have also been detected in the atmosphere [12,13].

In 2019, scientists from the Medical University of Vienna and the Austrian Federal Environment Agency detected the presence of microplastic particles in the feces of eight subjects participating in the study, and there were about 20 microplastic particles in every 10 g of feces on average [14]. In 2020, scientists in Italy first found microplastic particles in the placenta of pregnant women. Moreover, the scientists detected microplastics in both the fetal and maternal sides of the placenta, as well as in the developing membranes of the fetus. In this study, the scientists analyzed only about 4% of each placenta, and a dozen microplastic particles were detected, suggesting that the total number of microplastics could be much higher [15].

Although microplastics have been found in the human body, there has been no report on the potential harm of microplastics to the human body, including damage to the reproductive system. The reproductive system is very important for human health and reproduction, but the harm caused by microplastics to the reproductive system is not very clear. At present, the research on the damage to the reproductive system caused by microplastics is mainly focused on aquatic organisms [16] and mice [17], and the mechanism of the damage still requires further exploration.

Studies on aquatic biology have found that polystyrene microplastics significantly reduce the number and diameter of oocytes, decrease the fertilization rate and offspring larval production, slow development, and cause malformation of oysters [18]. Polystyrene microplastics (PS-MPSs) reduce the spawning quantity and hatching rate of Daphnia, as well as slowing the growth, changing the offspring sex ratio, and increasing the male proportion [19]. PS-MPSs can increase ovarian oxygen consumption and decrease the heart rate of zebrafish offspring [20]. After exposure to 10-micron polystyrene (2 μg/L, 20 μg/L, 200 μg/L) for 60 days, the ovarian pathology of medaka showed a decrease in mature follicles, an increase in early oocytes, a decrease in the volume of eggs and the fertilization and hatching rates, a reduction in estradiol and testosterone, and down-regulation of gonadotropin-releasing hormone, follicle-stimulating hormone, luteinizing hormone, follicle-stimulating hormone receptor, and luteinizing hormone receptor [21].

Regarding male reproductive development in mammalian mice, our laboratory and other researchers have consistently found that PS-MPSs can cause sperm and testicular damage in male mice. Exposure of BALB/C male mice to PS-MPSs resulted in a decrease in the number and loose arrangement of spermatogenic cells, a decrease in the sperm number, an increase in the malformation rate, a decrease in the activities of sperm metabo-lism-related enzymes (succinate dehydrogenase and lactate dehydrogenase), and a decrease in the serum testosterone content. Oxidative stress and activated JNK and P38 MAPK signaling pathways induce reproductive toxicity in mice [22]. Exposure of BALB/C male mice to PS-MPSs leads to reproductive toxicity through spermatogenesis disorder, decreased testosterone secretion, testicular tissue inflammation, and blood–testis barrier destruction [23]. Exposure of ICR male mice to PS-MPSs resulted in a decrease in the number of surviving sperm, an increase in the rate of sperm malformation and sperm cell atrophy, shedding, and apoptosis. Nf-κb and the inflammatory factors interleukins IL-1β and IL-6 were significantly increased, while the anti-inflammatory molecule Nrf2/HO-1 was decreased. Abnormalities in mouse spermatozoa were related to the Nrf2/HO-1/NF-κB pathway [24].

Mitochondria are intracellular energy factories that are also involved in the production of oxygen free radicals, the removal of misfolded proteins, and the regulation of cell death, which are of great significance for maintaining the normal physiological functions of cells [25]. The three key steps of spermatogenesis—mitosis, meiosis, and spermatogenesis—require the rearrangement of numerous organelles, including mitochondria, and the supply of energy. Spermatozoa lose most of their cytoplasm during differentiation, which further suggests an important biological reason for mitochondrial retention. Therefore, damage to the structure and function of mitochondria directly affects sperm quality [26]. Previous studies have shown that microplastic exposure can cause mitochondrial damage. After exposure to 200 μg/mL PS-MPSs, intracellular reactive oxygen species increased and the mitochondrial membrane potential decreased, leading to mitochondrial depolarization in human colon adenocarcinoma Caco-2 cells [27]. Studies on the exposure of BALB/C male mice to PS-MPSs showed that exposure to microplastics can lead to a decrease in the activity of succinate dehydrogenase, a marker enzyme in mouse sperm mitochondria, an increase in sperm ROS, and the induction of oxidative stress [22]. Exposure of the human alveolar epithelial A549 cell line to PS-NPs (25 nm, 70 nm) for 2 h resulted in up-regulation of the expression levels of the apoptotic proteins Bax, caspase-3, caspase-9, and cytochrome c in the mitochondrial pathway [28]. However, the role of mitochondria in microplastic-induced male reproductive injury has not been reported in the literature. Therefore, this study investigated the effect of mitochondria on microplastic-induced sperm injury in male mice and its mechanism.

This study conducted research on the reproductive toxicity of microplastics by constructing a mammalian model so as to clarify the potential health hazards of microplastics to the mammalian reproductive system and to explore the possible mechanism and whether the damage can be recovered in the natural state in order to provide a basis for further research. However, it has not been reported whether the damage to the mammalian male reproductive system caused by microplastics is reversible.

## 2. Materials and Methods

### 2.1. Microplastics

PS-MPSs with a particle size of 5 μm were purchased from Tianjin Bessile Chromatographic Technology Development Center (Tianjin, China) (see Appendix A for the particle characterization). Before the experiment, the mother liquor was treated with ultrasonic shock for 30 min. The PS-MPS solution used in the experiment was diluted proportionally with distilled water to the desired concentration.

### 2.2. Experimental Animals and Groups

Seven-week-old male ICR mice were purchased from Shandong Jinan Pengyue Laboratory Animal Breeding Co., Ltd. (Jinan, China). The mice were fed in an SPF animal room with a temperature of 22 ± 2 °C, humidity of 50–60%, and light/dark cycle of 12/12 h. All animals were fed adaptively for at least 7 days. Standard laboratory animal maintenance feed was purchased from Pengyue (Jinan, China) and water was freely provided. The study design is provided in Appendix A.

Animal model 1: Mice were randomly divided into six groups—three control groups and three exposure groups—with 10 mice in each group. The exposure dose of PS-MPSs was 10 mg/L. Mice in the control group drank water normally (without PS-MPSs). The daily water consumption per mouse was approximately 6–7 mL, and the average daily PS-MPS exposure per mouse was calculated to be 60–70 μg/day based on the amount of water the mice drank and the concentration of microplastics. According to the dietary habits of the mice, we changed the new PS-MPS solution at 9 p.m. every night and again at 9 a.m. the next day, shaking the water bottle at the right time to ensure that the mice could obtain enough PS-MPSs. Mice were exposed to PS-MPSs for 35 days. After the exposure, a control group and exposed group were randomly selected for related experiments, weighing the viscera weight and calculating the viscera coefficient at the same time. The remaining four groups received a normal diet. On day 71, a control group and an exposed group were randomly selected again for related experiments, weighing the viscera weight and calculating the viscera coefficient at the same time. Additionally, the remaining two groups continued on the normal diet. On day 106, the last two groups were selected for related experiments. At the same time, the weight of the organs was weighed and the organ coefficient was calculated. Animals were weighed before and after the experiment. Furthermore, animals were weighed and food intake calculated every 3 days until day 106.

Animal model 2: Mice were randomly divided into six groups and exposed for 0 w, 1 w, 2 w, 3 w, 4 w, and 5 w. The mice were sacrificed on days 0, 8, 15, 22, 29, and 36, and the relevant indexes were tested.

Animal model 3: Mice were randomly divided into five groups and exposed for 0 d, 1 d, 3 d, 5 d, and 7 d. Mice were sacrificed on days 0, 2, 4, 6, and 8 for relevant experiments.

### 2.3. Sperm Survival Rate, Motility and Deformity

In model 1, 6 mice were selected from the control group and the exposure group on day 36, day 71, and day 106. After the mice were sacrificed, the bilateral epididymis was removed, cut into pieces, and placed into TYH sperm capacitation fluid (A2050, Easycheck, Nanjing, China). After standing for 1–2 min, the sample was filtered with a 200-mesh filter, and the sperm survival rate and movement were detected on a machine (CFT-9202, Medrich, Xuzhou, China). The remaining samples were tested for sperm malformation under a microscope (BX63, Olympus, Tokyo, Japan) after Diff-Quick staining.

### 2.4. Testicular Histopathology

In model 1, mouse testicular tissues were stained with HE on days 36, 71, and 106 and observed under a microscope (BX63, Olympus, Tokyo, Japan).

### 2.5. WB Experiment

In model 1, testicular tissues were obtained from mice on days 36, 71, and 106. In model 2, testicular tissues were obtained from mice on days 0, 8, 15, 22, 29, and 36. In model 3, testicular tissues were obtained from mice on days 0, 2, 4, 6, and 8. The protein was extracted from 50 mg testicular tissue, and the protein concentration was quantified. The glass plate was then cleaned, aligned, and placed in the clamp; then, it was stuck vertically on the shelf to prepare for glue filling. Following this, 10% separating glue was mixed and shaken well immediately after adding TEMED for glue filling, and it was covered with anhydrous ethanol. After standing for 30 min, the gel solidified. Absolute ethanol was poured off and a 4% gum concentrate containing TEMED was filled in. The comb was then inserted into the gel concentrate, and after the gel concentrate solidified, the comb was pulled out, samples were added, and then SDS-PAGE electrophoresis was performed. After electrophoresis, the proteins on the gel were transferred to PVDF membranes. After transfer, blocking was performed with 5% nonfat milk powder, and then PVDF membranes were incubated with primary antibodies such as anti-PINK1 (1:1000; Affinity, DF7742, Changzhou, China), Parkin (1:1000; Affinity, AF0235, Changzhou, China), LC3B (1:1000; ABWAYS, CY5992, Shanghai, China), DRP1 (1:1000; Proteintech, 12957-1-AP, Wuhan, China), and OPA1 (1:1000; ABWAYS, CY7035, Shanghai, China) at 4 °C overnight. The next day, PVDF membranes were washed three times with TBST for 10 min each time. Horseradish peroxidase (HRP)-labeled secondary antibodies were then incubated at room temperature for 40 min and shaken slowly. PVDF membranes were cleaned with TBST three times for 10 min each time. After the photoluminescent solution was soaked, the bands on the film were displayed on a gel imaging instrument (5200 Multi, Tanon, Shanghai, China) and analyzed using Image J software.

### 2.6. Ordinary PCR and Long Fragment PCR

In model 1, genomic DNA was extracted from the control group and exposure group on days 36, 71, and 106. After DNA concentration and purity testing, DNA served as a template. The template was amplified through long-fragment PCR and ordinary PCR, and the primers were synthesized by Sangon Biotech in Shanghai. The long-fragment amplification system was the Long and Accurate PCR Kit (B639285, Sangon Biotech, Shanghai, China). The short fragment was amplified through ordinary PCR as an internal reference. Finally, gel electrophoresis was performed at a 2% agarose concentration, and the results were observed on a gel imager (5200 Multi, Tanon, China). Primer sequences are shown in Appendix A.

### 2.7. Real-Time PCR

In model 1, genomic DNA and RNA were extracted from the testes of the control group and the exposed group on days 36, 71, and 106. After the concentration and purity of DNA and RNA were detected, the RNA was reverted to cDNA with DNA and cDNA serving as templates. The primers required for real-time fluorescent PCR (PINK1, Parkin, OPA1, DRP1, COXI, and GAPDH) were synthesized by Shanghai biotechnology company. The reaction kit was the SYBR Green real-time fluorescent quantitative PCR kit (R602, NOBELAB, Beijing, China). The reaction system was configured according to the instructions, and the reaction conditions were set up. The reaction was carried out using a real-time fluorescent PCR instrument (Lightcycler 480Ⅱ, Roche, Basel, Switzerland). Primer sequences are shown in Appendix A.

### 2.8. Detection of ROS

An ROS assay was performed using the Reactive Oxygen Species Assay Kit (BB-470538, Bestbio, Shanghai, China) on days 36, 71, and 106 in model 1, on days 0, 8, 15, 22, 29, and 36 in model 2, and on days 0, 2, 4, 6, and 8 in model 3. Freshly obtained tissue samples were washed with PBS. An amount of 50 mg of tissue was accurately weighed, and 1 mL of homogenization buffer A was added; then, they were fully homogenized in a glass homogenizer. Centrifugation was conducted at 100× *g* for 3 min at 4 °C, and then the precipitate was discarded and the supernatant was obtained. Then, 200 μL homogenate supernatant and a 2 μL DHE probe were added to a 96-well plate and blown with a pipette to thoroughly mix them. This was followed by incubation at 37 °C for 30 min in the dark. The fluorescence intensity was measured at the excitation wavelength of 488 nm and emission wavelength of 610 nm. Another 50 μL supernatant homogenate was used for protein quantification, and the tissue ROS intensity was expressed as the fluorescence intensity/protein.

### 2.9. Detection of MDA

On days 36, 71, and 106 in model 1, 50 mg testicular tissue was placed into a 1.5 mL centrifuge tube, and 500 mL tissue lysate was added. A tissue homogenizer was then used for homogenization, with 10 s each time, for three times in total. After homogenization, it was centrifuged at 12,000× *g* for 10 min at 4 °C. The supernatant was carefully transferred to another centrifuge tube for further testing. Meanwhile, the protein content of the supernatant was also measured. The corresponding MDA detection solution and standard concentration were prepared according to specifications (S0131S, Beyotime, Shanghai, China). Then, 0.1 mL of homogenate, 0.1 mL of different concentrations of standards, and 0.1 mL of PBS were added into the black centrifuge tube as the control, followed by the addition of 0.2 mL of solution for MDA detection. The black centrifuge tube was placed in a metal bath at 100 °C for 15 min. When finished, it was cooled to room temperature and then centrifuged at 1000× *g* for 10 min. The absorbance at 532 nm was measured by a microplate reader after adding 200 µL supernatant into a 96-well plate. Finally, the MDA concentration was calculated according to the standard curve and then divided by the protein content. The content of MDA in testicular samples was obtained.

### 2.10. Detection of ATP

The ATP content of testicular tissue was detected using an ATP detection kit (S0026, Beyotime, Shanghai, China) on days 36, 71, and 106 in model 1 and on days 0, 8, 15, 22, 29, and 36 in model 2. An amount of 50 mg of testicular tissue was placed into a 1.5 mL centrifuge tube. Then, 500 μL ATP Lysate was added and homogenized in a tissue homogenizer for 10 s each time, for three times in total. After homogenization, it was centrifuged at 12,000× *g* for 5 min at 4 °C. The supernatant was carefully transferred to another centrifuge tube for further testing. The corresponding ATP standard concentration was prepared for the production of the standard curve according to specifications. The ATP detection reagent was diluted 1:9, added to a black 96-well plate, and left to stand for 4 min at room temperature. Then, 20 µL ATP standard and experimental samples were added into the orifice plate and detected by a multifunctional microplate reader.

### 2.11. Detection of Mitochondrial Membrane Potential

In model 1, on days 36, 71, and 106, the mitochondrial membrane potential was detected using a Rhodamine123 mitochondrial membrane potential detection kit (C2008S, Beyotime, Shanghai, China). Firstly, mitochondria in tissues were extracted with a tissue mitochondrial isolation kit (C3606, Beyotime, Shanghai, China). Then, 50 mg of tissue was cut and washed with PBS once. The tissue was placed in a centrifuge tube placed on ice, and scissors were used to cut the tissue into very small pieces. Five hundred volumes of precooled mitochondrial separation reagent A were added and homogenized in an ice bath for about 10 times. The homogenate was centrifuged at 600× *g* for 5 min at 4 °C. The supernatant was carefully transferred to another centrifuge tube and centrifuged at 11,000× *g* for 10 min at 4 °C. Then, the supernatant was carefully removed. The precipitate was the isolated mitochondria. Subsequently, the mitochondria were suspended again with mitochondrial storage reagents, and protein quantification was performed on the obtained mitochondria. Following this, an appropriate amount of Rhodamine 123 was taken (1000×), and then Rhodamine 123 was diluted according to the ratio of 1 mL detection buffer added to 1 µL Rhodamine 123 (1000×) and mixed well to form a Rhodamine 123 staining working solution. Mitochondria purified with a total protein content of 50 µg were added to 0.9 mL Rhodamine 123 staining working solution and detected by a fluorescent microplate reader. After mixing, a time scan was performed. The excitation wavelength was 507 nm, and the emission wavelength was 529 nm.

### 2.12. Statistical Analysis

SPSS 21.0 software (SPSS Inc., Chicago, IL, USA) was used for statistical analysis. The two groups of samples were compared using an independent *t*-test. When multiple groups were compared, the test of homogeneity of variance was performed first, and if the variance was even, a one-way ANOVA test was used, followed by Tukey’s multiple comparisons test. If the variance was uneven, a nonparametric test was used. Repeated measure analysis of variance was used to analyze the changes in the body weight and food intake of mice. When *p* < 0.05, the difference was considered statistically significant; otherwise, there was no statistical significance.

## 3. Results

### 3.1. General Condition of Exposed Animals during Exposure

As shown in Figure 1, during the 35 days of exposure, there was no statistically significant difference in the body weight, food intake, and organ weight or coefficient over time between the exposed and control groups. The organ weight and coefficient are shown in Appendix A.

### 3.2. Sperm Survival Rate, Motility, and Deformity during Exposure

As shown in Figure 2A–C, after exposure, compared with the control group, the sperm survival rate, forward motility, and malformation rate of mice in the exposure group were decreased, and the differences were statistically significant.

### 3.3. Testicular Pathology during Exposure Period

As shown in Figure 2D, through analysis of HE staining in the testicular tissue, the spermatogenic tubule cells in the exposed group were disordered, and spermatogenic cells at all levels were reduced.

### 3.4. Mitophagy during Exposure

As shown in Figure 3A, the protein contents of LC3BⅡ were decreased in the exposed group. Mitochondrial autophagy was also decreased.

### 3.5. Damage of Mitochondrial Structure and Function during Exposure Period

#### 3.5.1. Mitochondrial Autophagy Regulatory Proteins

As shown in Figure 3B, the protein expression of PINK1 and Parkin decreased in the exposed group, but the mRNA levels of the PINK1 and Parkin genes did not change compared with those in the control group, and the difference was not statistically significant.

#### 3.5.2. Mitochondrial Membrane Potential

As shown in Figure 3C, the mitochondrial membrane potential of exposed mice was significantly lower than that of the control group, and the difference was statistically significant.

#### 3.5.3. Mitochondrial Kinetic Homeostasis

As shown in Figure 3D, the expression of the mitochondrial fission protein Drp1 and fusion protein OPA1 decreased in the exposed group, and the difference was statistically significant, but the mRNA levels of the Drp1 and OPA1 genes did not change compared with those in the control group, and the difference was not statistically significant.

#### 3.5.4. Mitochondrial Genetic Characteristics

As shown in Figure 3E, compared with the control group, the mitochondrial gene integrity and copy numbers in the exposed group were decreased, and the differences were statistically significant.

#### 3.5.5. Mitochondrial ATP

As shown in Figure 3F, compared with the control group, the mitochondrial ATP content in the testicular tissue of the exposed group was significantly decreased.

### 3.6. Time Series of Mitochondrial Damage during Exposure

As shown in Figure 4A–C, mitochondrial damage proteins tended to decrease with time, but the PINK1 protein changed significantly. As shown in Figure 4D, the contents of ATP decreased gradually with the prolongation of the exposure time.

### 3.7. Oxidation Level of Testicular Tissue during Exposure Period

#### 3.7.1. ROS Levels

As shown in Figure 5A, the ROS level in the testicular tissue of the exposed group was significantly higher than that of the control group.

#### 3.7.2. MDA Level

As shown in Figure 5B, the MDA level in the testicular tissue of mice in the exposed group increased, and the difference was statistically significant compared with that in the control group.

#### 3.7.3. Relationship between Mitochondrial Structural Damage and Oxidation

As shown in Figure 5C, with the prolongation of the exposure time, the ROS level gradually increased, reached the highest level in the second week, and maintained this level. The expression level of the PINK1 protein reached the highest level on the fifth day of exposure and decreased one week later, and the difference was statistically significant.

#### 3.7.4. Relationship between Mitochondrial Function Damage and Oxidation

As shown in Figure 5D, with the prolongation of the exposure time, the ROS level gradually increased, reached the highest level in the second week, and maintained this level. The ATP content decreased and reached the lowest level in the fourth week.

### 3.8. General Conditions of Animals during Recovery

As shown in Figure 6, during the two recovery periods after the end of exposure, there was no statistically significant difference in the body weight, food intake, and organ weight or coefficient over time between the exposed and control groups. The organ weight and coefficient are shown in Appendix A.

### 3.9. Sperm Survival Rate, Motility and Deformity after Recovery

As shown in Figure 7A–C, after the first recovery period, the sperm survival rate of the exposed group was still lower than that of the control group, and the difference was statistically significant. The forward movement rate was lower than that of the control group, but the difference was not statistically significant. The deformity rate was higher than that of the control group, and the difference was statistically significant. At the end of the second recovery period, the sperm survival rate of the exposed group was lower than that of the control group, but the difference was not statistically significant. The forward movement rate was not different from that of the control group. The deformity rate was higher than that of the control group, but the difference was not statistically significant.

### 3.10. Testicular Pathology after Recovery

As shown in Figure 7D, at the end of the first recovery period, the testicular HE section staining showed that there was no difference in the spermatogenic tubules between the exposed and control groups.

### 3.11. Oxidative Levels of Testicular Tissue after Recovery

As shown in Figure 8A,B, at the end of the first recovery period, the levels of ROS and MDA in the testicular tissue were not significantly different from those in the control group.

### 3.12. Mitophagy after Recovery

As shown in Figure 8C, after the first recovery period, there was no significant difference in the LC3BⅡprotein level of testicular tissue between the exposed and control groups.

### 3.13. Mitochondrial Structure and Function after Recovery

#### 3.13.1. Regulatory Proteins of Mitophagy

As shown in Figure 8D, at the end of the first recovery period, the expression levels of the PINK1 and Parkin proteins in testicular tissues were not statistically different from those in the control group.

#### 3.13.2. Mitochondrial Membrane Potential

As shown in Figure 8E, after the first recovery period, there was no significant difference in the mitochondrial membrane potential of the testicular tissue between the exposed and control groups.

#### 3.13.3. Mitochondrial Kinetic Homeostasis

As shown in Figure 8F, after the first recovery period, there was no significant difference in the expression levels of Drp1 and OPA1 in the testicular tissue compared with those in the control group.

#### 3.13.4. Mitochondrial Genetic Characteristics

As shown in Figure 8G, after the first recovery period, there was no significant difference in the mitochondrial genome integrity and copy number of the testicular tissue between the exposed and control groups.

#### 3.13.5. Mitochondrial ATP

As shown in Figure 8H, after the first recovery period, there was no significant difference in the ATP content of the testicular tissue compared with that in the control group.

## 4. Discussion

Microplastics can damage the reproductive system of aquatic organisms and terrestrial organisms and threaten the survival and reproduction of biological species [29]. However, the mechanisms of damage to the reproductive system have not been fully understood and explained, and whether the effects of such damage are persistent or recoverable has not been reported. Therefore, our study is necessary.

The maturation of the mammalian male reproductive system requires a large amount of energy supply. Mitochondria, as the main source energy for cells and an important determinant of the sperm survival rate, play an important role in male reproduction [26]. Sperm development and the final fertilization both require the participation of mitochondria, whose damage can also affect the sperm quality and reproduction to a certain extent.

This research found that PS-MPSs could lead to mitochondrial autophagy in testicular tissue, which showed a trend of increasing first and then decreasing after exposure. In the initial stage, PINK1 and Parkin protein expression decreased while LC3BⅡ protein expression increased, indicating that it is possible that other autophagy pathways are activated besides the PINK1/Parkin pathway. In this stage, the damaged mitochondria are eliminated by autophagy so as to slow down the aggravation of tissue damage. The continuous exposure and the increased damage result in weaker mitophagy and aggravated damage to the structure and function of the testicular tissue.

The PINK1/Parkin mitophagy pathway is also a means of assessing mitochondrial quality [30]. Previous studies [31] have also found that mitophagy is closely related to sperm quality. Through a series of indexes of mitochondrial structure and function, the damage to mitochondria caused by polystyrene microplastics is variable. In this study, the reduced mitochondrial membrane potential disrupted the stability of the mitochondrial membrane and electron transport on the membrane, thus affecting its biological functions such as oxidative phosphorylation. It has been shown that the mitochondrial membrane potential affects the sperm survival rate [32]. Mitochondria divide and fuse continuously in the cell and maintain a certain proportion, which forms an intricate network system that maintains the structure and function of the cell [33]. In this study, both mitochondrial fission and fusion experienced a process of increasing first and then decreasing, and eventually both mitochondrial fission and fusion were damaged, resulting in the destruction of the structural integrity of the mitochondria. The results show a reduced mitochondrial DNA integrity and copy number, which are the mitochondrial genetic features closely related to sperm quality [34,35]. Due to the lack of an adequate protection mechanism, mitochondrial DNA is vulnerable to external influences and destruction. It also encodes a number of oxidative phosphorylation enzymes whose damage directly affects ATP production. Mitochondria make up about 15–22% of the total cell volume and provide 90% of its energy needs. Sperm travel through the mucus-filled cervix to the site of fertilization via the active flagella, whose energy is provided by the sperm’s mitochondrial oxidative phosphorylation, which is thought to be an important determinant of sperm motility. The sperm movement has a huge energy demand; thus, a reduced sperm vigor and concentration are two of the main reasons for male infertility. In this study, a decrease in the ATP content indicated a decrease in the sperm survival rate. ROS may be the cause of mitochondrial damage induced by PS-MPSs. The results show that ROS caused damage to the mitochondrial structure and its autophagy, led to abnormalities in the cell structure and function, and then caused abnormalities in the sperm in the testicular tissue. Previous studies have shown that oxidative stress can cause a decrease in the mitochondrial membrane potential [36], an imbalance in mitochondrial dynamic homeostasis [37], and oxidative damage to mitochondrial DNA [38]. Therefore, ROS induce damage to the structure and function of mitochondria in a certain range of exposure doses of PS-MPSs. However, with increasing doses, mitochondrial function is at a low level, suggesting that there may be a regulatory mechanism that keeps cells at a lower metabolic level, thus reducing the damage.

The reversibility of the reproductive toxicity of microplastics has not been reported in mammalian models, but the toxicity of other systems has been found to be irreversible in aquatic organisms. In one study, tilapia were exposed to microplastics in an aqueous environment. After 15 days of exposure and 15 days of recovery, hematological parameters did not change during the recovery period, except that the white blood cell count (WBC) and mean red blood cell volume (MCV) returned to normal. In addition, some parameters such as creatinine, uric acid, AST, ALT, and ALP were increased, which were still higher than those of the control group after the recovery period [39]. Studies have shown that the damage to the blood system caused by microplastics is irreversible.

In addition, some studies on the reversibility of the reproductive toxicity of nanomaterials have been reported. In a study on the reproductive toxicity of functionalized MWCNT-COOH in mice, it was found that carboxylated MWCNT-COOH caused a certain degree of oxidative stress and lipid peroxidation in testicular tissue under repeated exposure conditions, but these changes returned to normal in males of the 60-day and 90-day groups. Testicular histopathology showed similar changes [40]. In this study, sperm damage in mice improved after a recovery period but did not completely return to normal. Mitochondrial damage can reach normal levels after a recovery period. After two recovery periods of sperm injury, the index returned to the normal level.

It can be seen that the damage to sperm caused by PS-MPSs can return to the normal state after two spermatogenic cycles and under non-exposure conditions. It is suggested that the reproductive damage caused by PS-MPSs may be mainly induced by their particles because this damage can be reversed after the removal and effective elimination of the particles from the body, but only at experimental concentrations and in the form of exposure.

## 5. Conclusions

PS-MPS exposure resulted in a decreased sperm quality in ICR male mice. The decrease in sperm quality is related to the damage of mitochondria, and ROS may be an important cause of mitochondrial damage. The sperm damage induced by PS-MPSs was reversible at a certain concentration of exposure. Mitochondrial damage can be recovered after one cycle of spermatogenesis, while sperm damage needs two cycles of spermatogenesis to recover.

## Figures and Tables

**Figure 1 ijerph-19-14093-f001:**
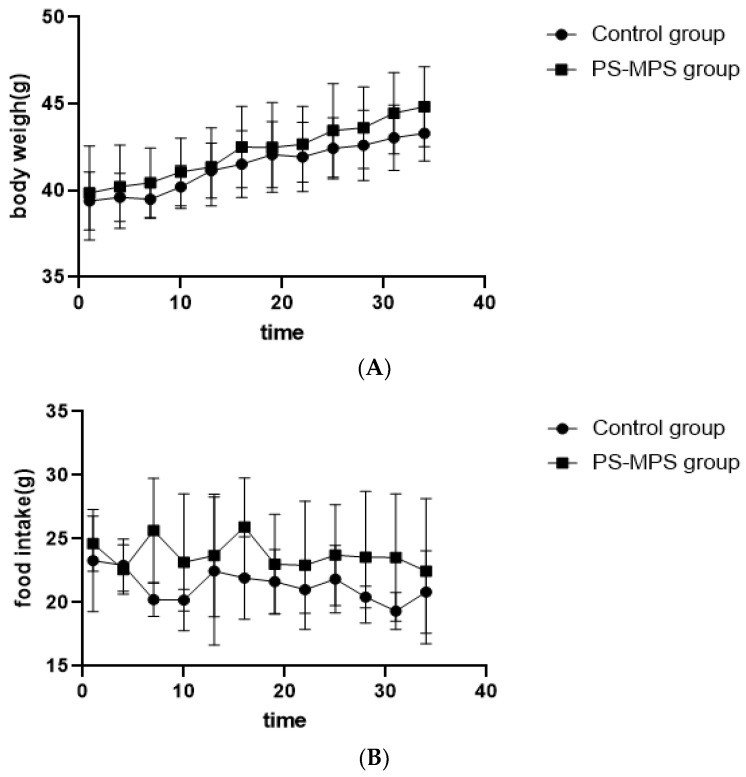
Body weight, food intake and organ coefficient of mice during the exposure period. (**A**) Body weight of mice; (**B**) Food intake of mice. The data are expressed as the mean ± SD (n = 8) and analyzed by dependent sample *t*-test.

**Figure 2 ijerph-19-14093-f002:**
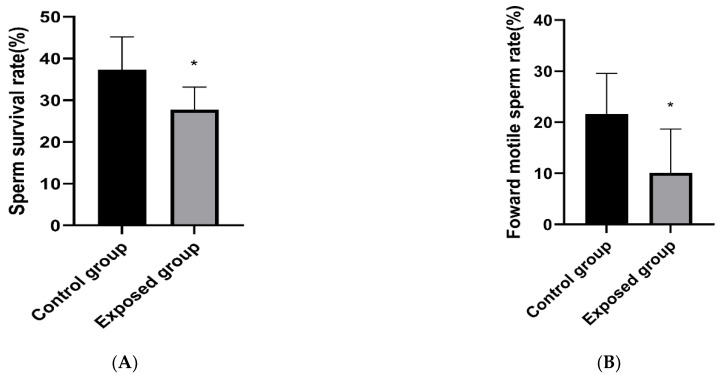
Sperm quality of mice and structure of the testis after exposure period. (**A**) Sperm survival; (**B**) Forward motion; (**C**) Deformity; (**D**) HE staining results of mice testicular tissue. The data are expressed as the mean ± SD (n = 8) and analyzed by dependent sample *t*-test. Note: * *p* < 0.05, indicates a statistically significant difference from the control.

**Figure 3 ijerph-19-14093-f003:**
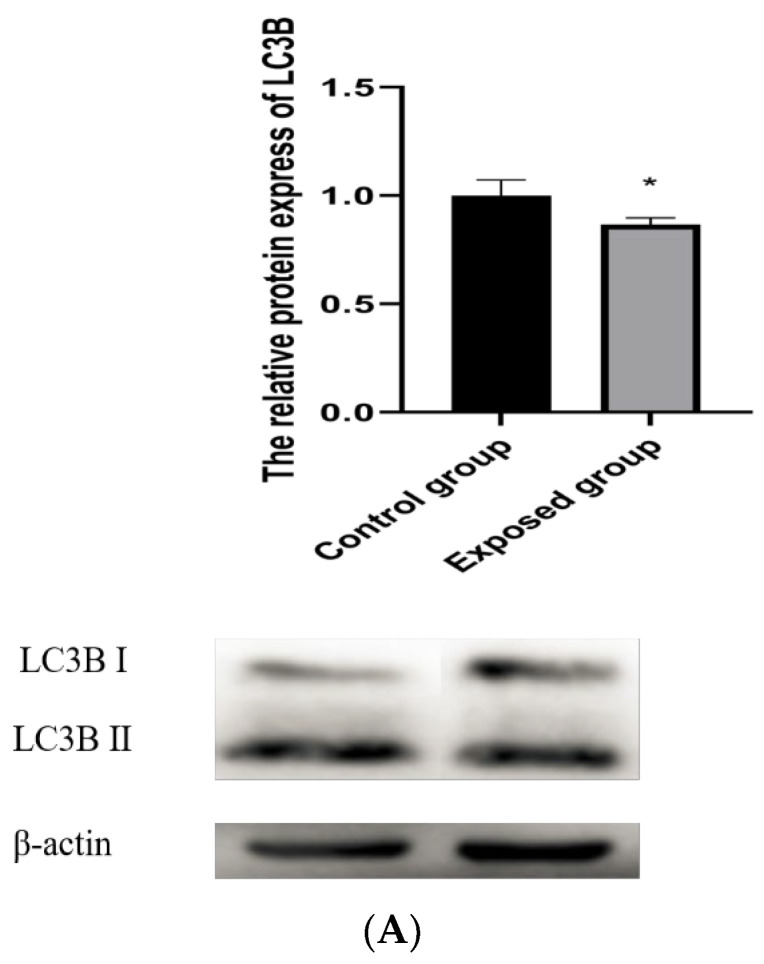
Mitochondrial structural and functional damage after exposure. (**A**) Expression of mitophagy protein; (**B**) Expression and transcription of mitophagy regulatory proteins; (**C**) mitochondrial membrane potential; (**D**) Expression and transcription of mitochondrial fission and fusion proteins; (**E**) Mitochondrial genetic characteristics; (**F**) ATP contents. The data are expressed as the mean ± SD (n = 8) and analyzed by dependent sample *t*-test. Note: * *p* < 0.05, indicates a statistically significant difference from the control.

**Figure 4 ijerph-19-14093-f004:**
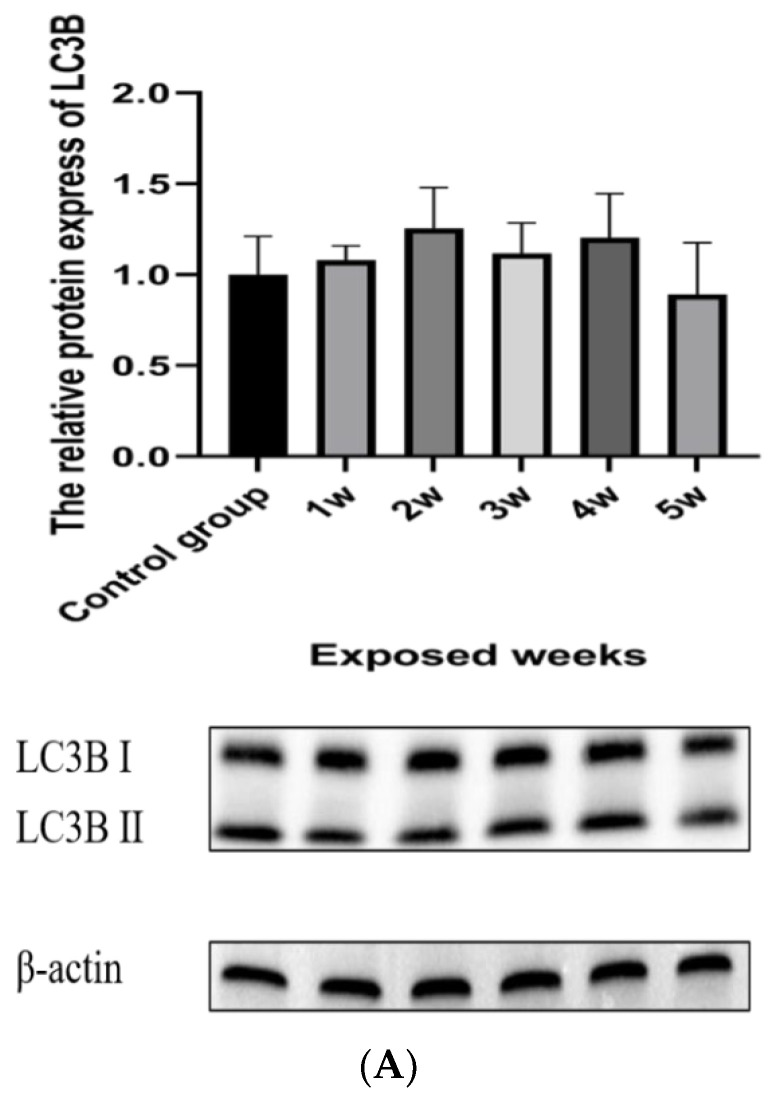
Time series of mitochondrial damage. (**A**) Mitophagy protein; (**B**) Mitophagy regulatory proteins; (**C**) Mitochondrial fission and fusion proteins; (**D**) ATP. The data are expressed as the mean ± SD (n = 8) and analyzed by ANOVA following the Turkey’s multiple comparisons test. Note: Lowercase letters are completely different, indicating significant difference (*p* < 0.05); any of the same lowercase letters means insignificant difference (*p* > 0.05).

**Figure 5 ijerph-19-14093-f005:**
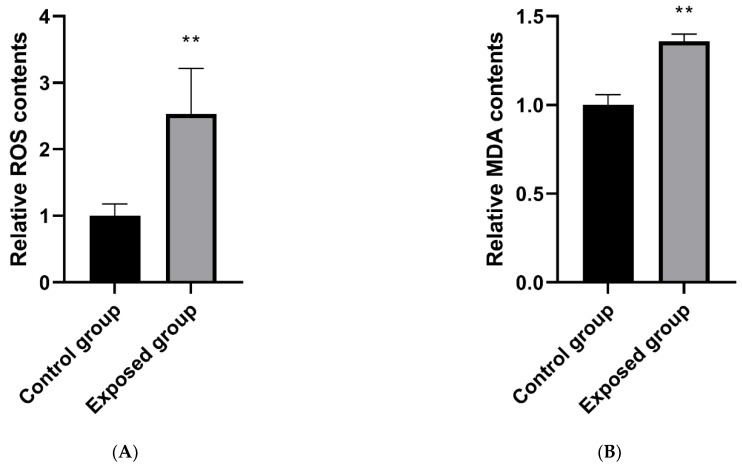
Oxidative levels in testicular tissue and relationship between mitochondrial damage and oxidation. (**A**) ROS level; (**B**) MDA level; (**C**) Mitochondrial structural damage and oxidation; (**D**) Mitochondrial function damage and oxidation. The data are expressed as the mean ± SD (n  =  8) and analyzed by dependent sample *t*-test or ANOVA following the Turkey’s multiple comparisons test. Note: ** *p* < 0.01 indicates a statistically significant difference from the control. Lowercase letters are completely different, indicating significant difference (*p* < 0.05); any of the same lowercase letters means insignificant difference (*p* > 0.05).

**Figure 6 ijerph-19-14093-f006:**
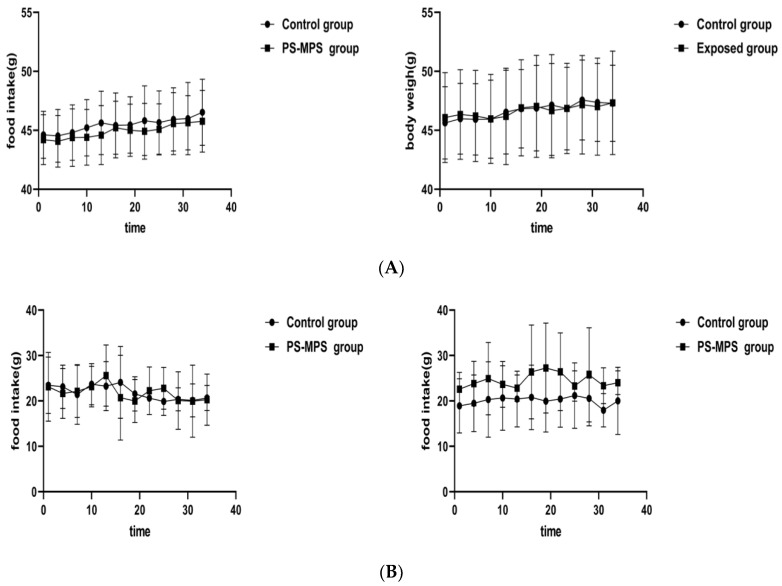
Body weight, food intake and organ coefficient of mice during the recovery period. The left side shows the change in the first recovery time and the right side shows the change in the second recovery time. (**A**) Body weight; (**B**) Food intake. The data are expressed as the mean ± SD (n = 8) and analyzed by dependent sample *t*-test. The left side of the figure shows the results of the first recovery period, and the right side shows the results of the second recovery period.

**Figure 7 ijerph-19-14093-f007:**
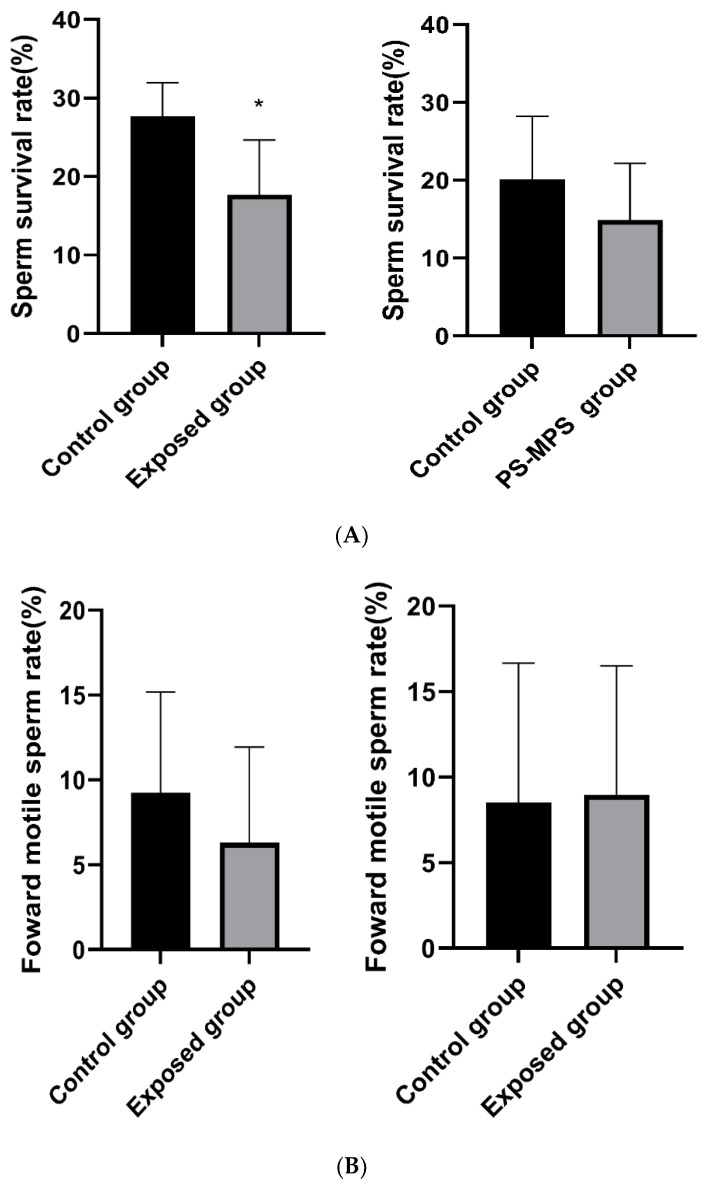
Sperm quality of mice and structure of the testis after recovery period. The left side shows the change after the first recovery time and the right side shows the change after the second recovery time. (**A**) Sperm survival; (**B**) Forward motion; (**C**) Deformity; (**D**) HE staining results of mouse testicular tissue after the first recovery time. The data are expressed as the mean ± SD (n = 8) and analyzed by dependent sample *t*-test. Note: * *p* < 0.05 indicates a statistically significant difference from the control. The left side of the figure (**A**–**C**) shows the results of the first recovery period, and the right side shows the results of the second recovery period.

**Figure 8 ijerph-19-14093-f008:**
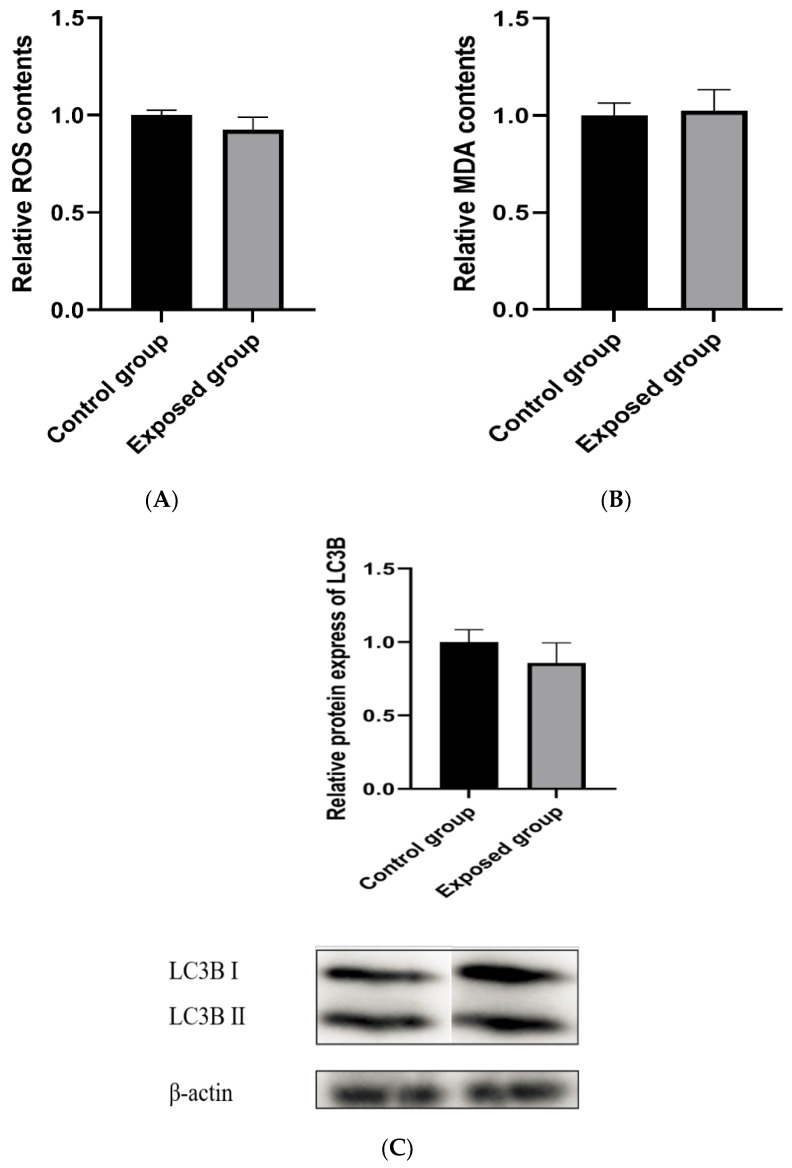
Oxidative levels of testicular tissue and mitochondrial structural and functional damage after the first recovery. (**A**) ROS level; (**B**) MDA level; (**C**) Expression of mitophagy protein; (**D**) Expression of mitophagy regulatory proteins; (**E**) Mitochondrial membrane potential; (**F**) Expression of mitochondrial fission and fusion proteins; (**G**) Mitochondrial genetic characteristics; (**H**) ATP contents. The data are expressed as the mean ± SD (n = 8) and analyzed by dependent sample *t*-test.

## Data Availability

Not applicable.

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
