# Peer review of "Determination of Biological and Molecular Attributes Related to Polystyrene Microplastic-Induced Reproductive Toxicity and Its Reversibility in Male Mice"

_ijerph, 2022, doi:10.3390/ijerph192114093_

Round 1
Reviewer 1 Report
Line 9: the term "human living environment" is not appropriate; "vegetables, drinking water, food" should not be listed together.
Line 51-56: No reference?
Line 57-62: No reference?
Line 93-100: No reference?
line 127: May need a table to present the study design and the timeline for each of the models, link the timeline to descriptions in the results.
The presentation of results was unreadable and unacceptable. Poor image quality, unclear labelling, explain exposure vs recovery timeline, and no explanation for observed changes and their corresponding exposure condition. No value+error in results was properly presented, and no supporting data was provided as supplementary material.
Line 449 - 466: No reference?
Reviewer 2 Report
16th September, 2022
Review of the Manuscript ID: ijerph-1936701, by T. Liu et al., entitled: “Reproductive toxicity and its reversibility of polystyrene microplastics” that is intended to be published as the Article in International Journal of Environmental Research and Public Health
(separate Microsoft Word file as Reviewer Attachment for Manuscript ID ijerph-1936701 Int. J. Environ. Res. Public Health 16th September 2022 that includes Comments to the Authors is also uploaded)
Considering research highlight, contribution of the Authors to the progress in the research area, thorough manner of data presentation, very well writing in English, as well as abundance of Results, Figures and Tables and relevant methods of statistical analysis, the quality of this paper deserves praise and merits my support. The Authors have received the high scores from me for the originality, importance of the work and the scientific value of their paper. In my opinion, the current paper provides insightful interpretation of topical and coming trends in thoroughly exploring the biological and molecular determinants of reproductive toxicity in male mice exposed to polystyrene microplastic-mediated endocrine disruptors. For those reasons, I strongly recommend the Editorial Board to allow for publication of this very interesting Article in International Journal of Environmental Research and Public Health, after the minor revision of the manuscript will have been completed by the Authors and provided that the Authors are ready to consider the Reviewer’s comments shown below:
1) The title of the manuscript should be re-edited to a more attractive and a more informative form as follows:
Determination of Biological and Molecular Attributes Related to Polystyrene Microplastic-Induced Reproductive Toxicity and Its Reversibility in Male Mice
2) There is a lack of the separate Abbreviations section in the paper. Therefore, this section should have been added to precisely elucidate and expand a wide range of in-text abbreviations, which have been used by the Authors in all the sections of their paper.
3) The References section has to be prepared in the format compatible with the requirements of International Journal of Environmental Research and Public Health.
General Comment of the Reviewer:
Before the manuscript will have been accepted for publication in International Journal of Environmental Research and Public Health, it requires the minor revision (according to the above-indicated remarks and recommendations of the Reviewer).

Reviewer 3 Report
I read with great interest the manuscript, which falls within the aim of this Journal. In my honest opinion, the topic is interesting enough to attract the readers’ attention. Nevertheless, the authors should clarify some points and improve the discussion, as suggested below.
Authors should consider the following recommendations:
- Manuscript should be further revised in order to correct some typos and improve style.
- I suggest to discuss, at least briefly, the potential correlation between endometriosis-associated infertility and endocrine disrupters such as 2,3,7,8-Tetrachlorodibenzo-p-dioxin (authors may refer to: PMID: 25920525; PMID: 31717614).
Author Response
Please see the attchment.

Round 2
Reviewer 1 Report
Thanks for responding to comments.
Please keep font consistent for all materials, specially figures.
Line 503: fish is auqatic organism, please check through the manuscript to avoid all these issues.
Author Response
Response to Reviewer 1 Comments
Point 1: Please keep font consistent for all materials, specially figures.
Response 1: Thank you for your valuable suggestion. We have harmonized the font throughout the article and are consistent with the figure.
Point 2: Line 503: fish is auqatic organism, please check through the manuscript to avoid all these issues.
Response 2: Thank you. We have corrected that statement. Now it becomes that Microplastics can damage the reproductive system of aquatic and terrestrial organisms, and threaten the survival and reproduction of biological species.